# Molecular Modelling of Ionic Liquids: Situations When Charge Scaling Seems Insufficient

**DOI:** 10.3390/molecules28020800

**Published:** 2023-01-13

**Authors:** Zhaoxi Sun, Lei Zheng, Zuo-Yuan Zhang, Yalong Cong, Mao Wang, Xiaohui Wang, Jingjing Yang, Zhirong Liu, Zhe Huai

**Affiliations:** 1College of Chemistry and Molecular Engineering, Peking University, Beijing 100871, China; 2NYU-ECNU Center for Computational Chemistry at NYU Shanghai, Shanghai 200062, China; 3Department of Chemistry, New York University, New York, NY 10003, USA; 4College of Physical Science and Technology, Yangzhou University, Yangzhou 225009, China; 5School of Chemistry and Molecular Engineering, East China Normal University, Shanghai 200062, China; 6NCS Testing Technology Co., Ltd., No. 13, Gaoliangqiao Xiejie, Beijing 100081, China; 7Beijing Leto Laboratories Co., Ltd., Beijing 100083, China; 8School of Environmental Science and Engineering, Suzhou University of Science and Technology, Suzhou 215009, China; 9XtalPi-AI Research Center, 7F, Tower A, Dongsheng Building, No.8, Zhongguancun East Road, Beijing 100083, China

**Keywords:** ionic liquids, partition coefficient, quinuclidinium, vdW scaling, force field

## Abstract

Charge scaling as an effective solution to the experiment–computation disagreement in molecular modelling of ionic liquids (ILs) could bring the computational results close to the experimental reference for various thermodynamic properties. According to the large-scale benchmark calculations of mass density, solvation, and water-ILs transfer-free energies in our series of papers, the charge-scaling factor of 0.8 serves as a near-optimal option generally applicable to most ILs, although a system-dependent parameter adjustment could be attempted for further improved performance. However, there are situations in which such a charge-scaling treatment would fail. Namely, charge scaling cannot really affect the simulation outcome, or minimally perturbs the results that are still far from the experimental value. In such situations, the vdW radius as an additional adjustable parameter is commonly tuned to minimize the experiment–calculation deviation. In the current work, considering two ILs from the quinuclidinium family, we investigate the impacts of this vdW-scaling treatment on the mass density and the solvation/partition thermodynamics in a fashion similar to our previous charge-scaling works, i.e., scanning the vdW-scaling factor and computing physical properties under these parameter sets. It is observed that the mass density exhibits a linear response to the vdW-scaling factor with slopes close to −1.8 g/mL. By further investigating a set of physiochemically relevant temperatures between 288 K and 348 K, we confirm the robustness of the vdW-scaling treatment in the estimation of bulk properties. The best vdW-scaling parameter for mass density would worsen the computation of solvation/partition thermodynamics, and a marginal decrease in the vdW-scaling factor is considered as an intermediate option balancing the reproductions of bulk properties and solvation thermodynamics. These observations could be understood in a way similar to the charge-scaling situation. i.e., overfitting some properties (e.g., mass density) would degrade the accuracy of the other properties (e.g., solvation free energies). Following this principle, the general guideline for applying this vdW-tuning protocol is by using values between the density-derived choice and the solvation/partition-derived solution. The charge and current vdW scaling treatments cover commonly encountered ILs, completing the protocol for accurate modelling of ILs with fixed-charge force fields.

## 1. Introduction

Ionic liquids (ILs) derivatives as greener non-flammable alternatives to traditional volatile organic solvents are becoming increasingly popular in various areas of modern chemical research, e.g., gas adsorption, refrigeration, adhesive, and electrochemical devices [1,2,3,4,5,6,7]. These novel solvents often have at least two components (e.g., ion pairs) with varying structural and chemical features and are thus highly tunable. They could be water-miscible or immiscible and possess high thermal stability and more importantly extraordinary solvation capability [8,9,10,11,12,13,14,15,16]. ILs derivatives have been applied as a non-aqueous media for chemical reactions, an extractant in extraction, an entrainer in distillation, and a sorbent for specific gas [17,18,19,20,21,22,23]. A critical application area of ILs in recent years is in drug delivery [24,25,26,27,28,29,30]. Many of the large-sized compounds with bioactivity have limited efficacy and side effects due to their poor aqueous solubility [31]. ILs as drug carriers and reservoirs could effectively enhance the solubility and permeability of these novel compounds and are incorporated into microemulsion carrier systems [32,33].

Molecular simulation as a powerful tool to probe the microscopic motions of different components in complex systems has been widely applied in various physical, chemical, and biological situations [34,35,36,37,38,39,40]. In atomistic simulations of ILs, fixed-charge force fields are often employed due to the cell size required to eliminate finite-size artifacts and the time scale needed to thoroughly sample the configurational space of relevance. Atomic charges are often obtained with some ab initio calculations due to the significant system dependence of these parameters, and the bonded and vdW parameters are often extracted from pre-fitted general-purpose parameter sets such as the general AMBER force field (GAFF) [41]. For highly charged species, the charge transfer and polarization effects could be quite significant and atomic charges derived from gas-phase calculations of isolated molecules produce too strong/intensive inter-molecular packing, triggering significant differences between the simulation outcome and the experimental observation [42,43,44,45,46]. A common solution to this problem is charge scaling [47,48,49,50,51,52,53], i.e., uniformly down-scaling/decreasing the atomic charges to improve the experiment–calculation consistency. Although the charge-scaling treatment is mathematically non-rigorous/sub-optimal, it does produce satisfactory results in many practical applications.

Observables commonly extracted from atomistic simulations of ILs derivatives are bulk properties, e.g., mass density, viscosity, and diffusion [54,55,56,57,58]. Consequently, these bulk properties are selected as the matching target of charge scaling. However, in our series of papers benchmarking the charge-scaling issue with large-scaling solvation free-energy calculations and also unbiased simulations of the bulk solvent, this density-matching procedure is found to degrade the reproduction of solvation free energies and water–ILs transfer free-energies [59,60], which is attributed to the overfitting of solvent–solvent interactions for the density-derived solution, while accurate calculation of solvation and partition thermodynamics requires balanced descriptions of solute–solvent and solvent–solvent interactions. Unlike bulk properties (e.g., mass density) that are monotonic with respect to the charge-scaling factor, the non-monotonic dependence of solvation/partition thermodynamics due to the competing electrostatic and vdW contributions of molecular solvation further complicates the situation [59,60]. From hundreds of solvation/partition data, we summarize that the scaling factor of 0.8 serves as a near-optimal solution generally applicable to most ILs [59,60], which is somehow consistent with electrostatic potential (ESP) analysis in isolated and small clusters [61]. Ion-pair-specific tuning of this parameter could still lead to marginal improvements, and the best solvation/partition-derived charge-scaling parameter is observed to be slightly larger than the density-derived solution [59,60]. Although this slight increase in the scaling factor could degrade the reproduction of the bulk property, it effectively improves the accuracy of solvation and partition thermodynamics and does not overfit a given observable. Thus, such treatment is considered to produce balanced descriptions of solvent–solvent and solute–solvent interactions. The large-scale benchmark on typical ILs and solute molecules of diverse structural and chemical features ensures the general applicability of the accumulated insights [59,60]. Direct application of the summarized modelling protocol to species significantly different from the benchmark set is attempted in the following work, specifically ILs involving Trihexyltetradecylphosphonium, L-Lactate, and (1S)-(+)-10-camphorsulfonate [62]. Interestingly, the 0.8-scaled restrained electrostatic potential (RESP) charges plus GAFF2 parameter combination is still able to accurately reproduce the mass density and the solvation/partition thermodynamics, validating the general applicability of the summarized general modelling guidelines. However, although the recommended modelling protocol has been proven suitable in many typical ILs systems, there are indeed outliers when the charge-scaling treatment seems unworkable. Specifically, either the charge-scaling treatment perturbs the computational outcome at a minimal level, or it could change the results to some extent; however, the simulation results are still far from the experimental reference using scaling factors in the normal range. In such situations, the vdW radius (particle size σ in the LJ potential or the so-called collision diameter) is often considered as an additional adjustable parameter, which would be tuned to improve the computation-experiment consistency [63,64,65,66,67].

Quinuclidinium and its derivatives as a family of bicyclic heterocycles with interesting electrochemical and pharmacological properties are crucial components in the ILs’ universe [68,69,70,71]. Both thermally and electrochemically, the aliphatic quaternary ammonium ions are more stable against oxidation and reduction than the imidazolium family [72]. The 1-alkylquinuclidinium cations are often associated with large organic fluorinated anions to form liquids at ambient conditions. A good candidate for this purpose is bis(trifluoromethylsulfonyl)imide [NTF], which is an anion studied extensively in our previous works [59,60]. The melting points of ionic solvents formed by 1-alkylquinuclidinium and NTF are critically influenced by the length of the aliphatic chain. For chains shorter than six carbon atoms, the compounds are all solids at the ambient temperature [72], while ILs involving longer chains such as 1-hexylquinuclidinium [QUIN6] and 1-octylquinuclidinium [QUIN8] are room-temperature ILs. In the current work, using the ILs formed by the two quinuclidinium cations and the [NTF] anion as examples of the failure of the charge-scaling treatment, we illustrate the impact of tuning vdW parameters on the widely considered bulk property, mass density, and the solvation and partition thermodynamics of a spectrum of solutes with diverse features, aiming at providing some general guidelines on parameter adjustments in situations when the charge-scaling treatment does not fully apply or seems insufficient to produce simulation outcome of sufficient accuracy.

## 2. Results and Discussions

### 2.1. Charge Quality from ESP Analysis

We first check the quality of atomic charges for ILs investigated in the current work as a pre-simulation parameter validation. By reproducing the ab initio ESP of individual molecules, the RESP charge scheme ensures the accurate calculation of inter-molecular electrostatic interactions with the Coulomb equation. Therefore, the quality of atomic charges is evaluated with the ESP reproduction. The ESP relative root-mean-squared error (RRMSE) is often computed with reference to the target of the regularized charge fitting, i.e., the HF/6-31G* ESP in the current case. It is often argued that this ab initio level would produce low-quality (inaccurate) results and shifting to higher-level Hamiltonians is preferred [73,74,75,76,77,78,79]. However, the necessity of replacing the traditional cheap choice HF/6-31G* with higher-level techniques is not really justified by the numerical data in practical ILs species. Previous papers in our ILs series already verified that the HF/6-31G* level serves as a reference level of satisfactory accuracy in typical ILs [59,62], but whether the conclusions could be generalized to the current quinuclidinium-based ILs remains unknown. Thus, we also computed the percentage deviations of the charge-produced ESP with respect to many other ab initio levels commonly employed in condensed-phase simulations and static chemical calculations, including B3LYP, BP86 [80,81], CAM-B3LYP [82], M05-2X [83], M06-2X [84], MN15 [85], PBE [86], PW6B95 [87], TPSSh [88,89], wB97X-D [90], B2PLYP [91], and mPW2PLYP [92]. To evaluate the performance of the crude Pople basis set 6-31G*, the basis set is also upgraded to def2-TZVPP and def2-QZVPP [93,94], which are sufficiently large in practical calculations. From the small magnitudes of ESP RRMSE with respect to the all gas-phase ab initio results depicted in Figure 1a–c, we confirm the absence of numerical problems in charge fitting for all ions forming ILs. Thus, the inapplicability of the traditional HF/6-31G* to charge generation of ILs claimed in many computational investigations is not really valid. A further argument about the level for ESP scan is the necessity of including solvent effects with some implicit-solvent models [73,74]. To benchmark this point, we re-scan the molecular ESP at all ab initio levels with IEFPCM [95] solvation (water parameters) and re-compute ESP RRMSE in Figure 1d–f. Numerical data clearly suggest that the gas-phase HF/6-31G*-targeted RESP charges produce the molecular ESP extremely close to all implicit solvent-involved levels. Therefore, the inclusion of solvent effects does not have a significant impact on charge generation. Further considering the fact that similar ESP reproductions have been proven in our previous work to result in similar bulk behaviors (e.g., mass densities) [53], we reach the conclusion that the traditional HF/6-31G* level is fully applicable to the charge derivation of ions forming ILs.

### 2.2. Density-Matching with Charge and vdW Scaling

We then perform molecular simulations and check the bulk property widely used as the criterion for parameter adjustment, the mass density. We first follow the normal charge-scaling protocol by scanning the mass density as a function of the scaling factor of atomic charges, the results of which are presented in Figure 2a. The experimental values of the mass densities obtained from literatures are 1.358 g/mL and 1.295 g/mL for [QUIN6][NTF] and [QUIN8][NTF], respectively [96]. It is clearly shown that the mass density does not show noticeable variations when the charge-scaling factor lies within the range of 0.6~0.9, which is in stark contrast to previous experiences in imidazolium-involved ILs. The mass densities are generally about 0.05 g/mL larger than the experimental values for both ILs. As this scanned 0.6~0.9 range is the normal and reasonable range of the charge-scaling factor employed in ILs simulations but the percentage deviation of mass density is still as large as ~4%, we conclude that the charge-scaling treatment cannot really bring the mass density (and more generally bulk properties) back to values of a satisfactory accuracy level and thus cannot effectively handle the problem. A common alternative in such cases is varying the vdW radius. As the percentage error of mass density is ~4% denser than the experiment, we scale up this tunable parameter uniformly with two scaling factors of 1.03 and 1.05 (i.e., 3% and 5% increases in the vdW radius), in order to loosen the vdW packing and thus create a bulk environment with a smaller mass density. The numerical data with both charge- and vdW-scaled parameter sets are also presented in Figure 2a. It is clear that the alteration of the vdW radius effectively tunes the bulk density, and the scaling factor of 1.03 performs best in minimizing the calculation-experiment deviation among the normal (100% GAFF2 parameters) and the two vdW-radius-enlarged parameter sets. A point worth noting is the minor ion-pair dependent behaviors in the vdW-radius scaling, although they are insignificant in the current cases, indicating the need for system-specific parameter adjustment in practical situations.

Further insights could be obtained by replotting the density profile along the vdW-scaling parameter, i.e., following the charge-scaling method. As the charge-scaling factor 0.8 is observed to produce satisfactory descriptions of bulk property and solvation thermodynamics and is considered as a near-optimal solution in our previous works [59,60,62] and also many others [56,61], we employed this charge scaling factor in the following investigations. The vdW-scaling-factor dependence is presented in Figure 2b. With the decrease in the vdW-scaling parameter, denser solvent packing is achieved, and the mass density of the bulk solvent increases monotonically. The 1.03 scaling parameter could be close to optimal for both ILs, and the 100% GAFF2 parameters produce slightly dense bulk packing. Although this 1.03 vdW-scaling parameter performs best in density-matching, it may not produce satisfactory results for solvation and partition thermodynamics, as the density-matching procedure only considers the solvent-solvent interaction, while the solvation and partition thermodynamics involve both solvent-solvent and solute-solvent interactions. Namely, the density-matching treatment could lead to a parameter set overfitting the bulk property and thus degrading the reproduction of the other non-targeted physiochemical properties. According to our previous experiences in charge scaling, the best solvation/partition-derived scaling factor is often slightly larger than the density-derived solution and produces denser solvent packing [59,60]. Following this guideline, in the current vdW-radius-scaling situation, the best scaling factor for vdW radius should be a bit smaller than 1.03, e.g., 102%. On the other hand, the 100% GAFF2 parameter set is unable to satisfactorily reproduce the solvent–solvent packing (mass density), but could reproduce solute–solvent interactions in a better manner. Therefore, what we expect to observe in the following large-scale fast-growth solvation free-energy calculations is that the normal GAFF2 vdW parameters would be able to produce solvation and partition thermodynamics with high accuracy, while the 103% σ set would generate larger errors for these observables. Another observation worth noting is that the response of mass density to vdW-radius scaling is almost linear. If we consider the packing of hard-sphere models, the response (derivative) of the system density in vdW scaling could be easily derived. However, due to the incorporation of many other potentials (e.g., torsional and electrostatic potentials), the behavior of the systems deviates from the model systems. Thus, we expect to summarize the system response by performing a linear regression for each vdW-scaling profile, which leads to two perfectly fitted curves (R^2^ ~0.999) with slopes −1.82 g/mL and −1.73 g/mL, respectively. The closeness of the two ~−1.8 g/mL slopes hints about the similar mechanisms of the vdW-scaling impacts on the bulk arrangement of quinuclidinium-based ILs. If we further transform the slopes into the relative variation by dividing them by the experimental mass densities, we reach the two almost identical values of −1.34 and −1.33 for the two ILs.

With a properly tuned force-field parameter set at 298 K, a further interesting investigation of the bulk density concerns its performance in the physiochemically relevant temperature range. To this aim, we extract further experimental values between 293 K and 343 K, and perform further unbiased simulations in the targeted temperature range. To make the temperature space more regular and include the existing 298 K data in this scan, the newly simulated temperatures consist of 288 K, 308 K, 318 K, 328 K, 338 K, and 348 K. Namely, the whole temperature scan is performed between 288 K and 348 K with a 10 K interval. The density profiles under the normal 100% GAFF2, 1.03-scaled, and 1.05-scaled parameter sets are presented in Appendix A. For the [QUIN6][NTF] ionic solvent, the 1.03 σ parameter set accurately reproduces the experimental densities in the whole temperature range, which confirms the fact that the vdW-scaling factor secured at 298 K could be generalized to neighboring temperatures. However, for [QUIN8][NTF], we observe a performance loss with an elevated temperature and the best vdW-scaling factor seems to be lying between 1.03 and 1.05. This phenomenon is not unexpected considering the space for further improvements via system-specific parameter adjustment suggested in Figure 2b, where the best vdW-scaling factor for this ionic solvent is found to be slightly larger than 1.03. As we are using the same vdW-scaling factor for the two ionic solvents and the 1.03 selection produces satisfactory results for both ILs at 298 K, we indeed believe it to be an acceptable option for the room-temperature investigation; however, further adjustments could be attempted when the simulation condition experiences critical changes.

Finally, it is worth comparing the current simulation outcomes with predictions provided by other techniques. A machine-learning predictor trained with more than 2000 experimental points was recently reported to achieve accurate predictions for mass density [97]. The predictions for the current two ILs [QUIN6][NTF] and [QUIN8][NTF] are also provided, which enables a face-to-face comparison with our simulation outcome. The 298 K results for the two ILs from the machine-learning predictor are 1.778 g/mL and 1.673 g/mL, both of which deviate significantly from the experimental references 1.358 g/mL and 1.295 g/mL. By contrast, our simulation estimates are 1.350 g/mL and 1.303 g/mL, which agree with the experiment satisfactorily. The success of the machine-learning predictor in many other ILs suggests it successfully captures the main-stream behaviors of the dataset, and the failure in the current quinuclidinium ILs should be related to the intrinsic behaviors of these species. This phenomenon is also in agreement with the modelling situation, where the widely applied charge-scaling treatment fails, and additional adjustments of vdW parameters are needed.

### 2.3. Predictive Power of Fine-Tuned Parameter Set on Solvation and Partition Thermodynamics

The check of mass density has starkly demonstrated the effectiveness of vdW-radius scaling in the molecular modelling of ILs. In this section, we investigate the impact of such a treatment on solvation and partition thermodynamics through large-scale nonequilibrium alchemical free-energy calculations. The atomic charges of ILs are scaled with the near-optimal solution 0.8 following our previous works [59,60,62], and the vdW-scaling factors employed include 1.00 (the normal GAFF2 parameters), 1.03 (the density-matching solution), and 1.05. Additionally, considering the computational settings of 1 ns per initial condition, 2.8 ns per pulling realization, 100 realizations per solute, and the solvation of 40 solutes in 2 ILs, the overall sampling time for the nonequilibrium alchemical free-energy calculation in this work hits 100 μs. As the cell size of ~10,000 atoms is quite large in modern solvation free-energy calculations, the computational endeavor for such a long-term sampling is considerable, making the current computational perspective comprehensive and solid.

The computational results and the experimental values [96,98,99,100,101] of solvation and partition thermodynamics are summarized in Appendix A. We first visualize the experiment–calculation correlation with the scatter plot in Figure 3. For solvation in both ILs, according to the number of outliers with large deviations (i.e., lying outside the ±1 kcal/mol and ±2 kcal/mol lines), the normal (100% GAFF2) parameter set seems best. The 103% σ set achieves an intermediate accuracy, and the 105% σ set has too many outliers. This performance rank agrees with the hypothesis discussed in the last paragraph of the previous section, which indicates that the vdW-scaling treatment does have behaviors similar to charge scaling considered in our previous works [59,60]. Combining the solvation-in-ILs results with the hydration data (producing the partition data) perturbs the scattering behavior to some extent, but the performance rank remains unchanged. Such obvious superiority of the normal 100% GAFF2 parameter set suggests its pronounced advantage. In the following paragraphs, we will present quality metrics from statistical analyses to provide a more quantitative evaluation.

To evaluate the closeness of the absolute values of thermodynamic observables, we compute the mean absolute error (MAE), the mean signed error (MSE), and the root-mean-squared error (RMSE), the values of which are also provided in Appendix A. For each of the three error metrics in both solvation-in-ILs and water-ILs-transfer cases, we observe the accuracy rank of 100% GAFF2 > 103% σ > 105% σ, which quantitatively confirms the superiority of the original parameter set. To compare the error sizes of the hydration case with the ILs-relevant situations, we present the bar plot with a mark of the hydration quality in Figure 4a. For solvation in ILs, both 100% GAFF2 and 103% σ could achieve acceptable accuracy levels. However, none of the three parameter sets produces satisfactory results for the water–[QUIN6][NTF] transfer, and only the 100% GAFF2 parameter set achieves the hydration accuracy for the water–[QUIN8][NTF] transfer. Thus, for the reproduction of absolute solvation and partition thermodynamics, the 100% GAFF2 parameter set is the best option among the three parameter sets. If the error size of the 103% σ is still acceptable, due to its better reproduction of the bulk property, this parameter set could also be used.

To evaluate the consistency of the ranking information from simulation and experiment, we compute Kendall’s rank correlation coefficient (τ) [102] and Pearlman’s predictive index (PI) [103], the values of which are also given in Appendix A and compared with the hydration case in Figure 4b. Obviously, the accuracy rank is the same as the RMSE situation. A behavior worth noting is the improved ranking coefficients when shifting from the solvation data to the partition thermodynamics, which should be related to error cancellations in the hydration and solvation-in-ILs data. Another correlation metric we compute is the Pearson *r*. From the numerical data in Appendix A, we identify good linear correlations between experiment and calculation for the normal GAFF2 parameter set, which are degraded to some extent upon the increase in the vdW-scaling factor.

Overall, the 100% GAFF2 parameter set achieves the best accuracy for solvation and partition thermodynamics. The 103% σ parameter set produces good solvation thermodynamics in both ILs, but the partition data have slightly larger errors. The 105% σ parameter set predicts solvation/partition thermodynamics with large errors and thus should not be used. Thus, the solvation- and partition-derived vdW-scaling factor is confirmed to be 100%. Combining the analysis of bulk property in the previous section with the current solvation/partition results, based on our previous experience in charge scaling, we recommend to use vdW-scaling factors between the density-derived choice 103% and the solvation/partition-derived solution 100%, as a balanced choice that simultaneously reproduces the bulk properties and solvation/partition thermodynamics with small errors. According to our previous charge-scaling experience, often scaling factors leading to slightly large mass densities are preferred. Therefore, in the current case, the balanced solution should be slightly smaller than 103%. Considering the sensitivity of thermodynamic observables to the variation of the vdW-scaling factor, values such as 102% should be the optimal choice for the current quinuclidinium-based ILs. Although it is preferable to derive some general scaling-factor applicable to most ILs, as the GAFF2 parameter set is pre-fitted without system-specific tuning such as the ab initio calculation-involved charge generation, the summarization of such a generally applicable vdW-scaling factor does not seem possible. However, the general guideline that the balanced vdW-scaling parameter should produce slightly larger mass densities than the experiment remains workable, which could guide ILs-specific vdW tuning in practical situations.

## 3. Computational Details

### 3.1. System Construction

The construction of the ILs system follows the previously accumulated guidelines in our previous works [59,60,62]. The ILs considered were formed with the cations [QUIN6] and [QUIN8] and the anionic [NTF], the 2D chemical structures of which are shown in Figure 5. While the two cationic species are structurally different from all ions modelled in our previous works but from a new family of ILs featuring the quinuclidinium core and alkyl chains of varying lengths on the quaternary ammonium center, the anion [NTF] was considered in several ILs modelled in our previous works. Thus, the previously accumulated experiences would have some applicability to the current [QUIN6][NTF] and [QUIN8][NTF] ILs, but some more system-specific parameter-tuning might be needed. The solute molecules as a probe in solvation and partition processes included 40 drug-like molecules with diverse chemical and structural features, and their 2D chemical structures are depicted in Figure 6.

The 3D chemical structures of all molecules except [QUIN8] were taken from the PubChem database, and that of [QUIN8] was obtained by modifying the aliphatic chain of [QUIN6] due to the absence of its structure in the database. The atomic charges were obtained with the RESP [104] charge scheme, i.e., B3LYP [105,106,107]/6-31G* optimization, HF/6-31G* ESP scan and regularized two-step fitting. After that, the atomic charges of ILs were scaled with some scaling factor to account for the charge transfer and polarization effects, the details of which are discussed later in the Results section. Atomic charges of solute molecules remained unchanged (i.e., the full RESP charges). The other parameters (e.g., bonded parameters and vdW radius) were obtained from the transferable GAFF2 [41] due to its good performance in previous applications [59,60,62]. The vdW parameters were also considered adjustable in the current work, the details of which will also be given in the Results section. For each ion-pair composition, we constructed a cubic cell with the PACKMOL package [108]. The number of atoms in each box was about 10,000 atoms, corresponding to ~170 ion pairs for both ILs. The threshold of 128 was benchmarked to be sufficient to eliminate finite-size artifacts in our previous work [59], and the current ~170 ion-pair setup was well above that threshold and thus would not cause noticeable finite-size problems. The cubic cell was replicated in the whole space with periodic boundary conditions. For water molecules, we used the TIP3P [109,110] model.

### 3.2. Molecular Simulation

For each pure ionic solvent box, we performed 5000 steps of geometry minimization, 10 ns NVT equilibration, and 20 ns NPT equilibration to reach the ambient condition (298 K and 1 atm, to be specified), after which a 30 ns NPT production run was initiated for density estimation. After that, the last snapshot was extracted as the starting configuration of the bulk ILs for alchemical free-energy calculation.

The alchemical method constructs an artificial pathway connecting physical end states to sample the transformation indirectly [111,112,113,114]. Equilibrium alchemical free-energy calculations are widely used to explore the conformational ensemble and compute thermodynamic information in chemical and biological systems [115,116,117]. In the context of molecular solvation, the method functions by gradually coupling or decoupling the solute–solvent interactions, the degree of which is regulated by the alchemical order parameter. However, a drawback is the need for equilibrium sampling in non-physical intermediate states, which can to some extent be avoided with the nonequilibrium technique [118,119,120,121,122]. The nonequilibrium free-energy technique deals with truly nonequilibrium data, e.g., the nonequilibrium work performed during the process. Simulations producing these data are often called steered molecular dynamics, where the system is governed by a time-dependent Hamiltonian instead of a static potential. The nonequilibrium sampling process is divided into two sub-steps. The first one is grabbing initial configurations/seeds, and the second one is performing nonequilibrium pulling simulations initiated from these initial conditions. Although there are approximated techniques to improve the efficiency, e.g., eliminating the need for initial configurational sampling with some selection criterion [123,124], in order to make the current simulation protocol fully bias-free and as general as possible, we retained the ordinary fast-switching protocol. Thus, equilibrium sampling at the solute–solvent decoupled state is performed to grab initial conditions, from which the unidirectional perturbation in the solute-creation direction is spawned. The 60 ns equilibrated solvent box mentioned above is merged with the gas-phase solute as the starting configuration of the decoupled state. The structure is energy-minimized 5000 steps and NVT-equilibrated for another 1 ns, after which a 100 ns production run with a sampling interval of 1 ns is performed to accumulate 100 independent initial seeds, from which nonequilibrium fast-switching transformations are spawned. The solute creation process is further divided into two sub-steps to avoid numerical instability. The first challenging vdW coupling step takes 2 ns, while the second solute charging step takes 0.8 ns due to its better convergence behavior. The variation of the alchemical order parameter is performed every 2 fs.

For post-simulation analysis, the microscopic works can be related to equilibrium thermodynamics with appropriate estimators, e.g., the Jarzynski’s scheme or the Crooks’ scheme [125,126,127]. Here, the free-energy difference during the solvation process is extracted with the asymptotically unbiased estimator exponential averaging (EXP), or equivalently the Jarzynski’s identity [125,126] in the context of nonequilibrium works, i.e., Δ*A*_solvation_ = − ln<exp(-*W*_growth_)> with *W_growth_* being the dimensionless microscopic work accumulated during the solute-creation transformation. To avoid underestimation of uncertainty due to the intrinsic behavior of the analytical formula [128], we perform bootstrap resampling to obtain the numerical uncertainty, which would be denoted by standard deviation (SD) in the following discussion.

In all simulations (i.e., brute-force simulation for density estimation and fast-growth transformation), the velocity rescaling algorithm [129] with a 0.2 ps time constant is employed for temperature regulation and the Parrinello–Rahman Lagrangian [130,131] with isotropic scaling and a 3 ps time constant is used for pressure regulation. Long-range electrostatics are treated with the smooth Particle-mesh Ewald [132] method and the cutoff for vdW interactions and the real-space electrostatics is set to 10 Å. All simulations are performed with GROMACS 2020.6 [133], while all other statistical analyses are performed with home-made codes.

## 4. Concluding Remarks

Molecular modelling is becoming a popular tool in the chemical research of green solvents. Fixed-charge additive force fields are commonly employed due to the time and length requirements of investigated processes. The force field selection of RESP+GAFF2 is found to be quite robust in modelling both the bulk properties and the interactions between ILs and external agents (solutes). As the full RESP charges do not properly take the polarization and charge transfer effects into consideration, it is common to scale down atomic charges via some property-matching procedure to minimize the experiment–calculation deviation. The bulk property widely selected as the matching target is the mass density. As the bulk-property matching procedure only considers the solvent–solvent interaction, often the best density-derived solution degrades the reproduction of the other component in the solution mixture, i.e., the solute–solvent interaction. Through large-scaling free-energy calculations for a spectrum of solute–solvent pairs and a detailed investigation of component-specific responses, we discover and elucidate the non-monotonic responding behavior of solvation and partition thermodynamics with respect to charge scaling and identify the scaling factor 0.8 as a solution generally applicable to typical ILs, producing a balanced description of solute–solvent and solvent–solvent interactions and thus simultaneously reproducing the experimental bulk property and solvation/partition thermodynamics. Although charge scaling as an effective treatment could handle the disagreement issue in many cases, there are situations when charge scaling seems insufficient to eliminate the systematic bias. In such cases, vdW parameters are often considered as another set of tunable parameters, which are varied to improve the experiment–calculation consistency.

The current work based on two ILs from the quinuclidinium family accumulates extensive numerical experiences in fine-tuning the vdW parameters (the radius parameter σ to be specific). The charge-scaling treatment cannot really perturb the mass density, and vdW-scaling effectively brings the computational results close to the experimental reference. Further, the best vdW-scaling factor obtained at 298 K is found to be effective in the physiochemically relevant temperature range between 288 K and 348 K, confirming the robustness of the vdW-scaling treatment. From the computation of solvation and partition thermodynamics under these vdW-scaled parameter sets, we notice that the vdW-scaling treatment has an impact similar to charge scaling. The improvement of mass density is often accompanied by the deterioration of the other non-targeted observables (e.g., solvation-free energies). Based on this observation, similar to the charge scaling case, we recommend a similar procedure of selecting the vdW-scaling factor, i.e., choosing a vdW-scaling factor between the density-derived choice and the solvation/partition-derived solution, in order to achieve a balanced description of different interactions involved in the complex solution system. The current vdW-scaling along with the previous charge-scaling treatment covers almost all commonly encountered ILs species, thus completing the general guidelines for accurate modelling with fixed-charge force fields.

## Figures and Tables

**Figure 1 molecules-28-00800-f001:**
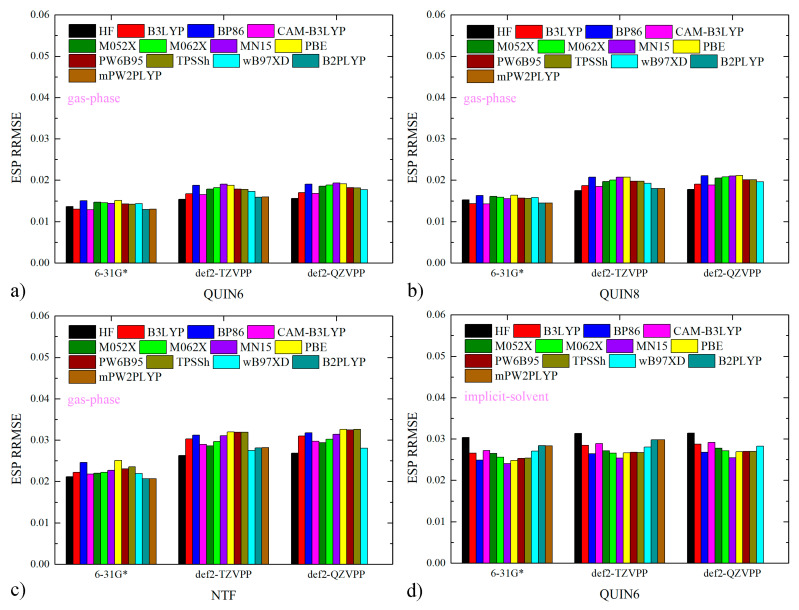
The percentage errors of charge-produced ESP from different ab initio references for the cations and anions forming ILs. The generation of the reference ESP data in (**a**–**c**) is performed in vacuo, while that in (**d**–**f**) is conducted in implicit solvent IEFPCM. Note that calculations at B2PLYP and mPW2PLYP levels with the def2-QZVPP basis set are not performed and thus the ESP deviations at these levels of theory are set at zero. Although the RESP charges are fitted with the gas-phase HF/6-31G* ESP, they perform quite well in reproducing the ESP data at the other higher levels. The inclusion of solvation effects via implicit solvent has little impact on this ESP reproduction, which supports the usage of the gas-phase HF/6-31G* level in charge generation.

**Figure 2 molecules-28-00800-f002:**
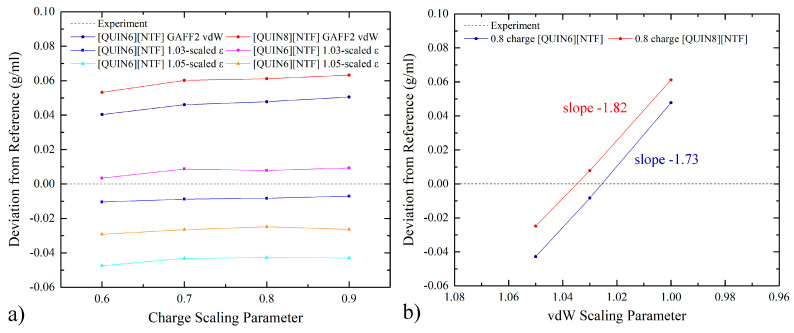
The deviation of the simulation-derived density from the experimental reference with different scaling factors for atomic charges and vdW radius (σ). The subplot (**a**) is generated along the normal charge-scaling variation, while the subplot (**b**) is drawn with respect to the magnitude of σ scaling. The statistical uncertainty of the simulated density is smaller than the point size. We perform a linear regression for each vdW-scaling profile and obtain two perfectly fitted (R^2^ ~0.999) curves with slopes −1.82 g/mL and −1.73 g/mL, respectively.

**Figure 3 molecules-28-00800-f003:**
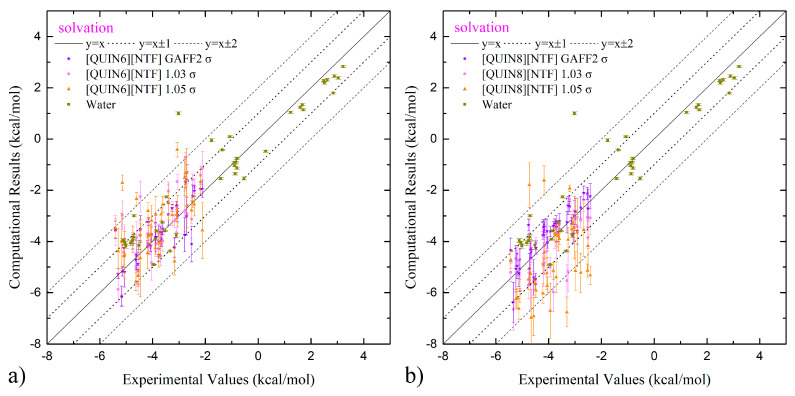
(**a**,**b**) The calculation–experiment correlation between the computed solvation free-energies in water and two ILs and the experimental reference and (**c**,**d**) that for water-ILs transfer free-energies with the normal GAFF2 vdW parameters, the 1.03-scaled-σ parameter set, and the 1.05-scaled-σ parameter set. The exact values of these free-energy estimates, the experimental values, and the quality assessment metrics are summarized in Appendix A.

**Figure 4 molecules-28-00800-f004:**
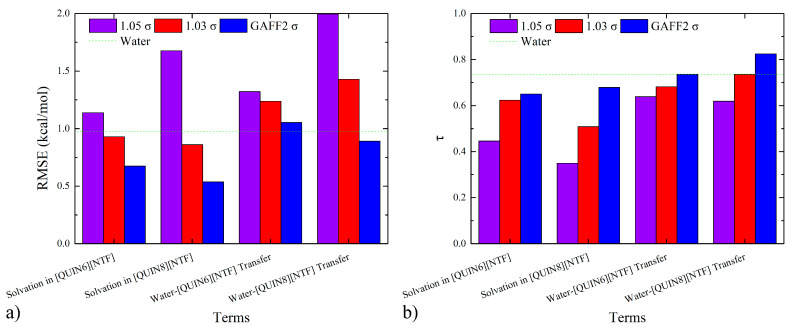
Quality metrics for infinite-dilution solvation-free energies in water and ILs and the water-ILs transfer free-energies under different vdW-radius scaling parameters: (**a**) the error metrics RMSE and (**b**) the ranking coefficient τ. The original GAFF2 vdW parameter set performs satisfactorily in reproducing the solvation and partition thermodynamics, and the vdW-scaling treatment leads to enlarged deviations, which is in agreement with the expectation that the density-matching procedure, although improving the description of solvent–solvent interactions, would degrade the balance between solute–solvent and solvent–solvent interactions.

**Figure 5 molecules-28-00800-f005:**
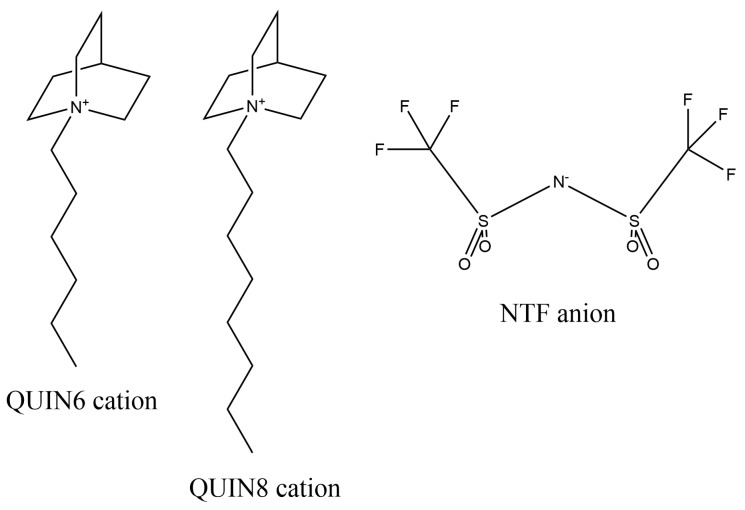
The ions forming the ILs investigated in this work. The two cationic species from the quinuclidinium family [QUIN6] and [QUIN8] are paired with the [NTF] anion to form two ILs of [QUIN6][NTF] and [QUIN8][NTF].

**Figure 6 molecules-28-00800-f006:**
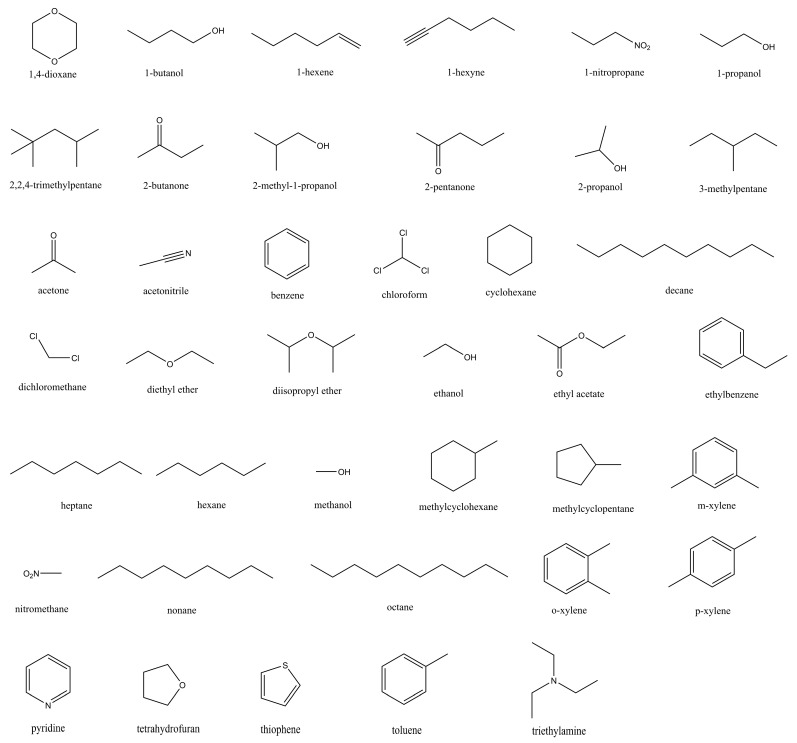
Solute molecules simulated in this work.

## Data Availability

The data that support the findings of this study are available from the corresponding author upon reasonable request.

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
