# Peer review of "Molecular Modelling of Ionic Liquids: Situations When Charge Scaling Seems Insufficient"

_molecules, 2023, doi:10.3390/molecules28020800_

Round 1
Reviewer 1 Report
In this manuscript, Sun et al reported the molecular modelling of Ionic liquids, this is a very noval research topic and the paper can be accepted after the following issue were concerned.
1. From Fig 5 we notice there are large simulation tolerance error between the computed solvation free energies and experimental references. However the trend between experiment and simulation matches well.
2. In the discussion section, can the authors comprare this work with others' machine learning work?
3. The novelty of this manuscript should further be highlighted.
4. What is the unique mechanism of this system, more dicussion should be added.5. Laterest references of this field can be added?
Author Response
Reviewer #1:
In this manuscript, Sun et al reported the molecular modelling of Ionic liquids, this is a very noval research topic and the paper can be accepted after the following issue were concerned.
- From Fig 5 we notice there are large simulation tolerance error between the computed solvation free energies and experimental references. However the trend between experiment and simulation matches well.
Response: We use error metrics to assess the closeness of the computational results and the experimental reference. Among error metrics, RMSE is sensitive to outliers and thus is investigated and discussed in detail, e.g., Fig. 6. For this statistical quantity, we do not see large values for solvation free energies, and the solvation in ionic liquids is comparable to the hydration case. Only for the 1.05 vdW radius set, RMSE grows to 1.1 kcal/mol (slightly larger than hydration). Therefore, overall, the calculation quality of solvation thermodynamics is satisfactory.
- In the discussion section, can the authors comprare this work with others' machine learning work?
Response: We added a comparison between the simulation outcome and the machine-learning predictions given by predictors published in Ind. Eng. Chem. Res. 2019, 58, 13, 5322–5338. Interestingly, although the machine-learning predictor is believed to be the state-of-the-art model currently, it fails with huge errors in the current quinuclidinium ILs. Some discussions about this issue are also added at the end of section 3.2.
- The novelty of this manuscript should further be highlighted.
Response: For most ionic liquids, the charge scaling treatment is sufficient for accurate calculation of various structural and thermodynamic observables, and some general rules for accurate modelling are summarized in our previous works and many others. However, there are situations that the charge scaling treatment cannot really solve the problem, and the simulation outcome cannot be brought close to experiment with charge scaling only. Therefore, in this paper, we propose to use the vdW scaling as a complementing solution, which could handle systems where charge scaling fails. The charge + vdW scaling treatments cover almost all ionic liquids and complete the general recommendations of accurate modelling summarized in our series papers. The title itself includes the information of insufficiency of charge scaling, the abstract gives details about charge and vdW scaling, and the results clearly validate the failure of charge scaling in [QUIN][NTF] ionic liquids and the success of the vdW scaling treatment.
- What is the unique mechanism of this system, more dicussion should be added.
Response: If using the hard-sphere packing model, we can easily derive the derivative of the density in vdW-radius scaling. However, due to the incorporation of many other potentials in the system (e.g., torsional and electrostatic potentials), the system exhibits more complex behaviors compared with simplified models. Thus, we have difficulties in providing a detailed description of the process, but we believe the slope of the density profile shown in Fig. 4b to capture some behaviors of this ionic-liquids family. Discussions about the above points are added to section 3.2 in the revision.
- Laterest references of this field can be added?
Response: We’ve added many latest references of the ionic-liquids field in the paper, including ref. 2-7, 47, 52-53, 59-60, 62-63, 109 (~20 related papers) published in 2022, ~15 related papers published in 2021, and many others published recently.
Reviewer 2 Report
The manuscript “Molecular Modelling of Ionic Liquids: Situations that Charge Scaling Seems Insufficient” by Zhaoxi Sun et al. touches the methodologically important question of parameter scaling in common force fields aimed at better representation of selected physicochemical properties of the studied systems. The approach is common in molecular modeling of ionic systems, such as ionic liquids (ILs) and deep eutectic solvents. In contrast to partial charge scaling, which is an established procedure in molecular dynamics (MD) simulations of ILs, the authors consider the van der Waals (vdW) radius of a particle as a tunable parameter, obtaining interesting results. Overall, I find this valuable manuscript publishable in Molecules, provided the authors respond to the comments listed below:
1) The authors state that “The number of atoms in each box is about 10,000 atoms, corresponding to ~170 ion pairs for both ILs, which is sufficiently large to eliminate finite-size artifacts.” It is actually well-known that IL systems are prone to finite-size effects, with the dynamic properties being more demanding in this respect (see J. Chem. Phys. 137 (2012) 094501) and systems of as many as 500 ion pairs are required to reach the linear scaling regime allowing extrapolation to infinite cell size. While the present work concerns static and thermodynamic properties only, some comment would be appropriate.
2) The authors mention the type of thermostat and barostat applied in the simulations, but omit the time constants, which can be important for reproducibility of computational protocol.
3) The authors do not mention the code used in the DFT charge fitting procedure. Also, what is the point of using the implicit solvent charge refitting if the authors essentially study pure IL systems? The solvent parameters used in IEFPCM are not detailed.
4) The caption to Figure 4 is way too long and repeats many details from the main text.
Author Response
Reviewer #2:
The manuscript “Molecular Modelling of Ionic Liquids: Situations that Charge Scaling Seems Insufficient” by Zhaoxi Sun et al. touches the methodologically important question of parameter scaling in common force fields aimed at better representation of selected physicochemical properties of the studied systems. The approach is common in molecular modeling of ionic systems, such as ionic liquids (ILs) and deep eutectic solvents. In contrast to partial charge scaling, which is an established procedure in molecular dynamics (MD) simulations of ILs, the authors consider the van der Waals (vdW) radius of a particle as a tunable parameter, obtaining interesting results. Overall, I find this valuable manuscript publishable in Molecules, provided the authors respond to the comments listed below:
1) The authors state that “The number of atoms in each box is about 10,000 atoms, corresponding to ~170 ion pairs for both ILs, which is sufficiently large to eliminate finite-size artifacts.” It is actually well-known that IL systems are prone to finite-size effects, with the dynamic properties being more demanding in this respect (see J. Chem. Phys. 137 (2012) 094501) and systems of as many as 500 ion pairs are required to reach the linear scaling regime allowing extrapolation to infinite cell size. While the present work concerns static and thermodynamic properties only, some comment would be appropriate.
Response: We added some clarifications about this ion-pair setup at the end of section 2.1.
2) The authors mention the type of thermostat and barostat applied in the simulations, but omit the time constants, which can be important for reproducibility of computational protocol.
Response: The values of time constants are added in the revision.
3) The authors do not mention the code used in the DFT charge fitting procedure. Also, what is the point of using the implicit solvent charge refitting if the authors essentially study pure IL systems? The solvent parameters used in IEFPCM are not detailed.
Response: We use home-made codes for RESP fitting. The procedure follows the classic antechamber implementation. As for the implicit-solvent treatment, the IEFPCM model is used to account for solvation-induced polarization, which would lead to differences in molecular ESP. The use of implicit solvent models in charge generation is also recommended in many recent papers, e.g., the RESP2 paper 10.1038/s42004-020-0291-4. The implicit solvation IEFPCM uses water parameters. This detail is added to the end of section 3.1 in the revised manuscript.
4) The caption to Figure 4 is way too long and repeats many details from the main text.
Response: The discussions in the caption of Figure 4 are removed in revision.